# Pesticide Residues and Heavy Metals in Vineyard Soils of the Karst and Istria

**Borut Vrščaj [1,2,\*]**, **Helena Baša Česnik [3]**, **Špela Velikonja Bolta [3]**, **Sanja Radeka [4]** **and Klemen Lisjak [5]**

1 Agricultural Institute of Slovenia, Department of Agroecology and Natural Resources, Hacquetova ulica 17, SI 1000 Ljubljana, Slovenia
2 Faculty of Environmental Protection, Trg mladosti 7, SI 3320 Velenje, Slovenia
3 Agricultural Institute of Slovenia, Central Laboratories, Hacquetova ulica 17, SI 1000 Ljubljana, Slovenia
4 Institute of Agriculture and Tourism, Department of Agriculture and Nutrition, Karla Huguesa 8, 52440 Poreč, Croatia
5 Agricultural Institute of Slovenia, Department of Fruit Growing, Viticulture and Oenology, Hacquetova ulica 17, SI 1000 Ljubljana, Slovenia
\* Correspondence: borut.vrscaj@kis.si

**Abstract:** Pesticide residues and heavy metal concentrations were determined in two depths (0–20 and 20–40 cm) of Chromic Cambisol in 69 vineyards in the Slovenian winegrowing region of the Karst. Similarly, pesticide residues and heavy metal concentrations were also determined in two depths of Calcaric Cambisol in 11 vineyards in the other Slovenian and Croatian winegrowing regions of Istria. The topsoil (0–20 cm) was analysed for the presence of 176 different pesticidal active substances using two multiresidue analytical methods: (a) gas chromatography coupled with mass spectrometry (GC/MS) and (b) liquid chromatography coupled with tandem mass spectrometry (LC/MS/MS). Seven active substances (five fungicides and two insecticides) were detected in the soil samples. Their concentrations were compared with the maximum concentrations observed in the vineyards of the winegrowing regions of France, Italy, and Spain. In addition to pesticides, the soil samples were analysed for the presence of nine heavy metals commonly detected in vineyard soils. The concentrations of arsenic, cadmium, cobalt, chromium, copper, molybdenum, nickel, lead, and zinc were below the critical thresholds set by Slovenian legislation, with the exception of one soil sample in which the Cu concentration exceeded the critical threshold. Compared with the maximum concentrations measured in other vineyard soils in Spain, Italy, and France, the heavy metal concentrations in the vineyard soils of Karst and Istria were lower. Both the heavy metal concentrations and the residual concentrations of pesticidal active substances in the vineyard soils of the Karst and Istria regions were significantly lower than the concentrations that are occasionally discussed in the literature.

**Keywords:** contamination; chromic cambisol; calcaric cambisol





## 1. Introduction

Grapes are an important and widespread crop in the Mediterranean region of Europe. Fresh and dry grapes, juices, and wines are an important part of the human diet in some countries. The use of pesticides is considered conventional agricultural practice in viticulture [1]. The intensity of grapevine protection measures depends on the characteristics of the grapevine pests and viticultural practices (conventional, integrated, or organic) [2]. Navarro et al. [3] reported that grape berry moths (*Eupoecilia ambiguella* and *Lobesia botrana*), powdery mildew (*Uncinula necator*), downy mildew (*Plasmopara viticola*), and grey mold (*Botrytis cinerea*) are the most common insects/diseases affecting vines. In Slovenia, 133 different plant protection products (PPPs), with 67 different active substances, are authorised for wine grape production. Of these, 35 are fungicides [4]. Sixteen of these fungicides contain some form of Cu and 26 contain some form of Zn. The conventional application



of pesticides (spraying) is not effective for certain species. The conventional application of pesticides could lead to overuse and the significant spread of pesticides to non-target plants, soils, water, and air, thus polluting the environment. Due to its properties, soil itself plays an important role in the fate of pesticides. It is considered to be an important sink for pesticide residues, and at the same time, it is a source of pesticide residues; indeed, pesticides often enter the air or aquatic environments as a result of these sinks [5]. Furthermore, increased pesticide concentrations in soil can affect soil biodiversity, as they can significantly alter soil microbial communities [6].

Commonly used agricultural practices, often referred to as traditional or intensive farming practices, are a cause of concern for consumers and environmentalists as a result of the contamination of vineyard topsoil (0–20 cm) with pesticides; therefore, monitoring pesticide residues in vineyard soils and other agricultural soils should be deemed an important environmental protection measure that provides necessary information for the development of, or introduction of, agricultural practices that can maintain soil health and soil biodiversity at an appropriate level. This should help prevent the unacceptable contamination of soil, groundwater, surface water, and air with pesticide residues.

The presence of heavy metals (HMs) in soil can be either natural or anthropogenic in origin. The weathering of parent material (rock) that is rich in HMs can contribute to elevated or even high concentrations of individual HMs in soils. The anthropogenic sources of the contaminants are mainly industry, transport, and often agriculture. The most common sources of HMs are fertilisers (especially cadmium (Cd) and lead (Pb)), pesticides (especially copper (Cu), mercury (Hg), manganese (Mn), lead (Pb), zinc (Zn), and arsenic (As)), and the manure or sewage sludge that is applied to agricultural soils (especially Cd, chromium (Cr), nickel (Ni), selenium (Se), and molybdenum (Mo)) [7]. The mobility and bioavailability of HMs depend on a number of soil properties (the content of clay and organic matter in the soil) as well as on the physical (porosity, soil structure, permeability) and chemical conditions in the soils (acidity and cation exchange capacity, soil moisture, etc.). The soil acts as a matrix and as a perfect trap for successfully retaining HMs, especially if it is rich in clay and organic matter; therefore, the leaching of HMs and their uptake by plants is usually relatively low compared with the total amount of HMs in the soil. As a result, HMs gradually accumulate in the (top) soil profile over long periods of time [8], and they can reach concentrations that are harmful and pose a threat to the soil biota, plants, humans, and the environment. Due to the gradual accumulation and persistence of HMs in the soil, monitoring HMs in vineyard soil is even more important than monitoring pesticide residues.

Viticulture and wine production are economically important activities in many agricultural areas of the world. In general, winegrowers from large and recognised wine-producing countries in Europe (such as France, Italy, Spain, etc.) often operate large-scale farms and own extensive estates and wineries.

In terms of economic resilience, the economy of small winegrowers cannot be compared with such large winegrowers. The average vineyard area of farms in Slovenia has increased slightly in recent years, amounting to an increase of 0.55 ha [9]. In addition to their own production, winegrowers also buy grapes from neighbouring farms. The average production of the most well-known wineries in Slovenia is 15,000 hectolitres per winery. Economically, the small wineries are strongly dependent on the national market.

Information on (low levels of) HMs and pesticide residues in soils and wine can be important for marketing, brand promotion, and organic production, which is becoming increasingly recognised in the region. In addition to economic factors, sustainable agricultural production, soil health, and reduced pesticide use are becoming increasingly concerning for producers. On the other hand, consumers, who are frequently faced with unbiased and generalised public information on soils that are polluted with HMs and pesticide residues, which impact food as well as wine, are becoming increasingly interested in soil health, quality, and the nutritional value of food; therefore, they tend to welcome scientific information on wine quality.

The aim of this paper is to present the findings of a case-study investigation of the concentration of pesticides and HMs in the soils of vineyards in the winegrowing regions of Karst and Mediterranean Istria (Figure 1).

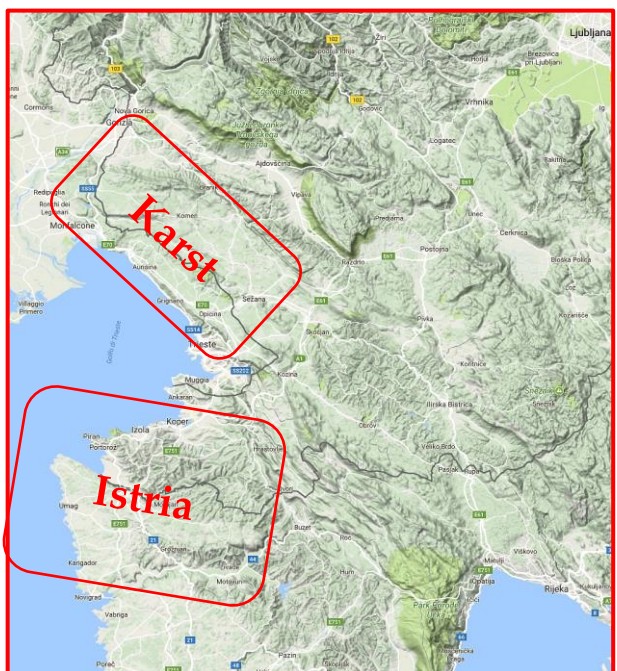 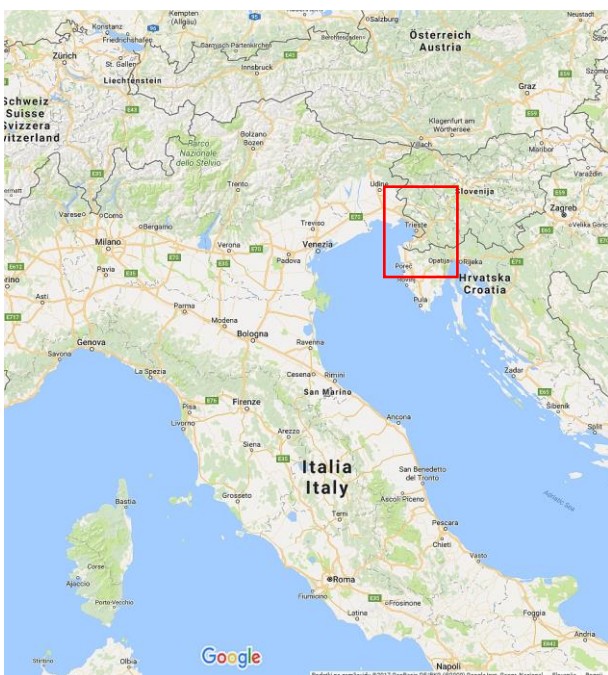

**Figure 1.** The Karst region (Slovenia) and the cross-border winegrowing regions of Istria (Slovenia/Croatia) (Source: Google).

Although these measurements were taken for other purposes in 2012 and 2015, the soil and grape contamination data are interesting to compare with general opinions on vineyard soil quality and to support wine producers in the cross-border Karst and Istria winegrowing regions.

## 2. Materials and Methods

### 2.1. Soil Sampling

In spring 2012, a total of 69 soil samples were taken from two depths (0–20 cm and 20–40 cm) in 35 Karst vineyards. In spring 2015, the topsoil of 11 vineyards in the winegrowing region of Istria was sampled at a depth of 0–20 cm. Five samples were taken on the Slovenian side of the border and six were taken on the Croatian side of the same cross-border winegrowing region. The locations of the soil samples in the vineyards were selected in accordance with how well they could best represent natural conditions (primarily taking into account: predominant soil type, location in the central part of the vineyard, distance from possible sources of contamination and roads, and the representative slope for the vineyard). The soils were sampled with HM-free tools (e.g., a wooden spatula and scraper).

### 2.2. Pesticide Residue Analyses

The extraction was carried out with a mixture of acetone, petroleum ether, and dichloromethane in a ratio of 1:2:2. The solvent mixture was added to the air dried and ground soil samples. The samples were shaken for 2 h. The extracts were then filtered, evaporated, and analysed by gas chromatography coupled with mass spectrometry (GC/MS), and liquid chromatography coupled with tandem mass spectrometry (LC/MS/MS). See Appendix A for the list of 86 active substances.

*2.3. The Pseudo Total HM Content Analyses*

The soil samples were air dried and ground to particles smaller than 250 μm. Aqueous extracts were prepared from the samples using a mixture of hydrochloric and nitric acid (Aqua regia). The samples were then allowed to stand at room temperature for 16 h and they were subsequently boiled under reflux for 2 h. The extracts were filtered and made up to the required volume with diluted nitric acid. The individual HMs were determined using atomic absorption spectrometry (AAS). The concentrations of Cu, Zn, Ni, and Cr were measured by flame AAS, whereas As, Cd, Co, Mo, and Pb were measured using electrothermal AAS.

## 3. Results and Discussion

*3.1. The Concentrations of Pesticide Residues in Topsoil*

**Seven active substances** were found in 80 soil samples from Karst and Istria, mainly in the upper soil layer (0–20 cm). Considering the fact that most infections in grapes are caused by fungi, coupled with the relatively large number of fungicides that were registered for grape cultivation in Slovenia (35), and the average number of PPP applications per growing season (at least 10 times) [10], it is not surprising that five of the seven active substances found were fungicides and only two were insecticides (13 insecticides are registered for grape cultivation in Slovenia). No active substances from herbicides were found in the samples.

The five most important vineyard pests treated by the seven active substances detected in the soil samples are as follows:

- Powdery mildew (*Uncinula necator*): boscalid, quinoxyfen, tetraconazole;
- Grey mould (*Botrytis cinerea*): boscalid;
- Downy mildew (*Plasmopara viticola*): dimethomorph, chlorothalonil;
- Moths (Eupoecilia ambiguella, Lobesia botrana): chlorphyriphos; and
- Leafhopper (*Scaphoideus titanus*), which is controlled by chlorphyriphos.

The most frequently found active substance was chlorphyriphos, which was the only active substance identified in the Istria region. The concentrations of the identified active substances can be found in Table 1.

**DDT-based pesticides** have not been used in vineyards since the 1970s [11]; however, some residues were found in Karst soil, which is probably due to their high concentration in soil [12]. The concentration of DDT was far below the critical threshold value of 4 mg kg$^{-1}$ [13]; this is the legal threshold value, above which, the soil is considered unsuitable for growing food and feed without risk of adverse health effects on humans and animals. The results are shown in Table 1.

No pesticide residues were detected in the topsoil of 63.8% of samples (44) from the Karst winegrowing region, and no pesticide residues were found in 81.8% of the samples (9) from the Istria winegrowing region. According to these results, pesticide residues in these winegrowing regions pose little or no risk to the environment or to agricultural production.

The results of the measurements were additionally compared with the concentrations of pesticide residues found in vineyards in France and Spain. Schreck et al. (2008) analysed 54 soil samples from a vineyard in Gaillac (France), which were taken throughout the entire growing season, from April to October, before, during, and after the application of pesticides. During the growing season, six analytes could be easily detected. The insecticides, chlorphyriphos and lambda-cyhalothrin, were the only ones detected in topsoil samples (0–5 cm depth). The concentrations of the two insecticides were very low and were at the limit of detection (0.005 and 0.001 mg kg$^{-1}$, respectively). These results suggest that chlorpyriphos was retained in the soil and not degraded. Metalaxyl and myclobutanil (fungicides) were the two studied compounds that were most present in the soil. They were detected in the lower soil layers, to a depth of 40 cm, mainly due to their water solubility, and their highest concentration was 0.003 mg kg$^{-1}$. In the topsoil, the concentration of myclobutanil reached 0.008 mg kg$^{-1}$, and for metalaxyl, the concentration

reached 0.05 mg kg$^{-1}$ in one exceptional case—one day after pesticide application. The herbicides, flumioxazin and flazasulfuron, were applied directly to the soil surface, and they were detected at a soil depth of 0–20 cm in concentrations ranging from 0.01 mg kg$^{-1}$ to 0.025 mg kg$^{-1}$.

**Table 1.** Pesticide residues in vineyard soil samples.

| Active Substance | | LOQ * (mg kg$^{-1}$) | Region | Depth (cm) | Concentration (mg kg$^{-1}$) | No. of Samples ** | CTV *** (mg kg$^{-1}$) |
|---|---|---|---|---|---|---|---|
| Boscalid | fungicide | 0.004 | Karst | 0–20 | 0.008 | 1 | / |
| Boscalid | fungicide | 0.004 | Karst | 20–40 | <0.004 | / | / |
| Chlorothalonil | fungicide | 0.001 | Karst | 0–20 | 0.001–0.004 | 3 | / |
| Chlorothalonil | fungicide | 0.001 | Karst | 20–40 | <0.001 | / | / |
| Chlorpyriphos | insecticide | 0.002 | Karst | 0–20 | 0.002–0.018 | 12 | / |
| Chlorpyriphos | insecticide | 0.002 | Karst | 20–40 | 0.002–0.006 | 5 | / |
| Chlorpyriphos | insecticide | 0.002 | Istria | 0–20 | 0.004–0.006 | 2 | / |
| DDT | insecticide | 0.01 | Karst | 0–20 | <0.01 | / | 4 |
| DDT | insecticide | 0.01 | Karst | 20–40 | 0.03 | 1 | 4 |
| Dimethomorph | fungicide | 0.001 | Karst | 0–20 | 0.001–0.005 | 5 | / |
| Dimethomorph | fungicide | 0.001 | Karst | 20–40 | 0.002 | 3 | / |
| Quinoxyfen | fungicide | 0.002 | Karst | 0–20 | 0.002–0.004 | 3 | / |
| Quinoxyfen | fungicide | 0.002 | Karst | 20–40 | 0.002 | 1 | / |
| Tetraconazole | fungicide | 0.002 | Karst | 0–20 | 0.002 | 1 | / |
| Tetraconazole | fungicide | 0.002 | Karst | 20–40 | <0.002 | / | / |

* LOQ: Limit of quantification; ** Number of samples with quantified residues; *** CTV: Critical threshold value.

Bermúdez-Couso et al. [14] analysed 28 soil samples from vineyards in Galicia (Spain) in March 2005, March 2006, and May 2006. The vineyard soils contained the following fungicides: cyprodinil at a concentration of 0.003–0.078 mg kg$^{-1}$, fludioxonil at a concentration of 0.008–0.295 mg kg$^{-1}$, metalaxyl at a concentration of 0.002–0.221 mg kg$^{-1}$, penconazole at a concentration of 0.003–0.411 mg kg$^{-1}$, and procymidone at a concentration of 0.003–1.124 mg kg$^{-1}$.

Pose-Juan et al. (2015) analysed 17 soil samples from vineyards in La Rioja (Spain) in March 2012. Soil from the vineyards contained acetochlor (an herbicide), at levels ranging from 0.005 to 0.1 mg kg$^{-1}$; fluometuron (an herbicide), at levels ranging from 0.005 to > 0.1 mg kg$^{-1}$; hydroxyterbuthylazine (an herbicide), at levels ranging from 0.005 to 0.1 mg kg$^{-1}$; kresoxime—methyl (a fungicide), in quantities up to 0.005 mg kg$^{-1}$; methoxy-fenozide (an insecticide), in quantities up to 0.005 mg kg$^{-1}$; and terbuthylazine (an herbicide), in quantities ranging from 0.005 to > 0.1 mg kg$^{-1}$.

The Karst and Istria regions belong to the Mediterranean part of Europe; therefore, a comparison with southern France and Spain is meaningful. We can conclude that in Karst and Spain, fungicides are the most frequently found pesticide residues in vineyard soil. In France, fungicides are found in similar concentrations to insecticides and herbicides. In Istria, only one insecticide was found, which was the same insecticide found in France. Pesticide residue levels reached up to 0.05 mg kg$^{-1}$ in France and 1.124 mg kg$^{-1}$ in Spain; these levels are higher than those found in Karst and Istria.

### 3.2. Concentrations of Pesticide Residues in Grapes and Wine

Despite the considerable amount of pesticides used in viticulture, the yielded end products in Karst and Istria (i.e., grapes and wine) were obtained in lower quantities than expected. Moreover, they are considered safe for consumers in terms of possible

food contamination. In the Karst region [15], pesticide residues in grapes were below the maximum residue limit (MRL). More specifically, 59% of all grape samples contained residue amounts that were 10% under the MRL, with regard to the active substances in question. Similarly, no pesticide residues were detected at all in 33% of all wines from Teran PTP. In the Istria region [16], pesticide residues were 10% below the MRL in all of the grape samples analysed. No pesticide residues were detected in 71% of all Malvazija wine samples from Istria.

### 3.3. Heavy Metals in Vineyard Soil in Karst and Istria

3.3.1. The Recent National Legislation

The HM concentrations that were determined in the soil samples of the vineyards were interpreted on the basis of the relevant Slovenian national legislation [13]. The cited regulation is based on three main thresholds: the limiting threshold value (LTV), the warning threshold value (WTV), and the critical threshold value (CTV). The threshold values presented in Table 2 can be interpreted as follows:

**Table 2.** Slovenian HM concentration threshold values in soil (RS, 1996), in mg kg$^{-1}$, using air dried soil.

|  | As | Cd | Co | Cr | Cu | Mo | Ni | Pb | Zn |
|---|---|---|---|---|---|---|---|---|---|
| **Limit threshold value (LTV)** | 20 | 1 | 20 | 100 | 60 | 10 | 50 | 85 | 200 |
| **Warning threshold value (WTV)** | 30 | 2 | 50 | 150 | 100 | 40 | 70 | 100 | 300 |
| **Critical threshold value (CTV)** | 55 | 12 | 240 | 380 | 300 | 200 | 210 | 530 | 720 |

The **LTV** is the concentration of a single hazardous substance in soil that does not adversely affect plant growth conditions and is not harmful to animals; it does not lower groundwater quality or soil fertility. When the concentration is below the LTV, the effects of HMs in soil on human health and the environment are acceptable. Soil is considered unpolluted if: (a) the concentration of a single HM exceeds the natural background but remains under the LTV, or (b) any other man-made hazardous substance considered by the regulation [13], is elevated but below the LTV. In such cases, soil deserves attention, and—if or where possible—protective measures. Unfortunately, the protection measures are not defined by regulations.

The **WTV** is the concentration of a single hazardous substance in soil that may pose a risk to human or animal health or adversely affect the environment in certain types of land use. If the concentration of a single hazardous substance exceeds the WTV, the soil is considered polluted.

The **CTV** is the concentration of a single hazardous substance in soil that may pose a serious threat to humans, animals, and the environment in general. Soil containing a hazardous substance above the CTV is considered unsuitable for the general agricultural production of food and feed; it poses a risk of contamination to groundwater and significantly degrades the quality of the environment.

3.3.2. Heavy Metal Concentrations in Vineyard Soils in Karst and Istria

The analytical data on the heavy metal concentrations in the total of 80 soil samples were statistically evaluated. The statistical parameters for the individual heavy metal concentrations in the soils and the proportion of heavy metal concentrations exceeding the LTV, WTV and CTV limits are shown in Tables 3–5.

**Table 3.** HMs in vineyard topsoil (0–20 cm) in the Karst region in spring 2012 (*n* = 35).

| | As | Cd | Co | Cr | Cu | Mo | Ni | Pb | Zn |
|---|---|---|---|---|---|---|---|---|---|
| Minimum concentration (mg kg$^{-1}$) | 13.9 | 0.10 | 12.3 | 70.6 | 35.0 | 2.0 | 39.7 | 29.1 | 73.8 |
| Maximum concentration (mg kg$^{-1}$) | 33.2 | 4.1 | 41.0 | 145 | 304 | 10.8 | 102 | 162 | 197 |
| Average concentration (mg kg$^{-1}$) | 19.7 | 0.9 | 24.1 | 94.4 | 95.3 | 5.2 | 66.9 | 43.0 | 103 |
| Median concentration (mg kg$^{-1}$) | 19.1 | 0.6 | 23.8 | 87.9 | 60.6 | 4.8 | 68.6 | 37.5 | 98 |
| STDEV (mg kg$^{-1}$) | 4.6 | 0.8 | 5.4 | 18.5 | 69.2 | 2.2 | 14.4 | 24.1 | 24.1 |
| Portion of samples < LTV (%) | 62.9 | 68.6 | 20.0 | 65.7 | 48.6 | 94.3 | 8.6 | 94.3 | 100.0 |
| Share of samples ≥ LTV and <WTV (%) | 31.4 | 22.9 | 80.0 | 34.3 | 17.1 | 5.7 | 54.3 | 0.0 | 0.0 |
| Share of samples ≥ WTV and <CTV (%) | 5.7 | 8.6 | 0.0 | 0.0 | 31.4 | 0.0 | 37.1 | 5.7 | 0.0 |
| Share of samples ≥ CTV (%) | 0.0 | 0.0 | 0.0 | 0.0 | 2.9 | 0.0 | 0.0 | 0.0 | 0.0 |

**Table 4.** HMs in the vineyard soil layer at 20–40 cm in the Karst region in spring 2012 (*n* = 34).

| | As | Cd | Co | Cr | Cu | Mo | Ni | Pb | Zn |
|---|---|---|---|---|---|---|---|---|---|
| Minimum concentration (mg kg$^{-1}$) | 13.9 | 0.2 | 14.9 | 49.9 | 33.0 | 1.8 | 40.8 | 28.9 | 67.3 |
| Maximum concentration (mg kg$^{-1}$) | 35.3 | 3.0 | 40.7 | 133 | 270 | 14.4 | 102 | 133 | 175 |
| Average concentration (mg kg$^{-1}$) | 20.0 | 0.8 | 23.8 | 93.0 | 87.3 | 5.2 | 66.7 | 40.3 | 98.9 |
| Median concentration (mg kg$^{-1}$) | 18.7 | 0.6 | 22.0 | 87.1 | 58.6 | 4.7 | 67.6 | 37.4 | 94.6 |
| STDEV (mg kg$^{-1}$) | 5.0 | 0.6 | 5.6 | 31.7 | 66.4 | 2.6 | 14.7 | 17.3 | 21.0 |
| Share of samples < LTV (%) | 61.8 | 73.5 | 23.5 | 64.7 | 52.9 | 94.1 | 11.8 | 97.1 | 100.0 |
| Share of samples ≥ LTV and <WTV (%) | 32.4 | 23.5 | 76.5 | 35.3 | 20.6 | 5.9 | 47.1 | 0.0 | 0.0 |
| Share of samples ≥ WTV and <CTV (%) | 5.9 | 2.9 | 0.0 | 0.0 | 26.5 | 0.0 | 41.2 | 2.9 | 0.0 |
| Share of samples ≥ CTV (%) | 0.0 | 0.0 | 0.0 | 0.0 | 0.0 | 0.0 | 0.0 | 0.0 | 0.0 |

**Table 5.** HMs in the vineyard topsoil at 0–20 cm in the Istria region in spring 2015 (*n* = 11).

| | As | Cd | Co | Cr | Cu | Mo | Ni | Pb | Zn |
|---|---|---|---|---|---|---|---|---|---|
| Minimum concentration (mg kg$^{-1}$) | 4.3 | 0.16 | 14.6 | 37.5 | 28.2 | 0.1 | 48.2 | 12.2 | 57.8 |
| Maximum concentration (mg kg$^{-1}$) | 27.2 | 0.58 | 42.8 | 178 | 96.7 | 13.4 | 102 | 40.5 | 88.3 |
| Average concentration (mg kg$^{-1}$) | 13.4 | 0.3 | 22.6 | 86.5 | 55.6 | 2.9 | 73.1 | 26.8 | 70.3 |
| Median concentration (mg kg$^{-1}$) | 8.8 | 0.2 | 22.8 | 83.3 | 54.4 | 0.3 | 66.9 | 28.3 | 66.2 |
| STDEV (mg kg$^{-1}$) | 8.6 | 0.1 | 7.8 | 38.4 | 19.5 | 4.8 | 18.5 | 11.8 | 9.8 |
| Share of samples < LTV (%) | 63.6 | 100.0 | 36.4 | 63.6 | 72.7 | 81.8 | 9.1 | 100.0 | 100.0 |
| Share of samples ≥ LTV and <WTV (%) | 36.4 | 0.0 | 63.6 | 27.3 | 27.3 | 18.2 | 45.5 | 0.0 | 0.0 |
| Share of samples ≥ WTV and <CTV (%) | 0.0 | 0.0 | 0.0 | 9.1 | 0.0 | 0.0 | 45.5 | 0.0 | 0.0 |
| **Share of samples ≥ CTV (%)** | **0.0** | **0.0** | **0.0** | **0.0** | **0.0** | **0.0** | **0.0** | **0.0** | **0.0** |

### 3.3.3. Concentrations of Individual Heavy Metals in Vineyard Soil

The Chromic Cambisol soil types of the Karst region have developed on Middle and Upper Cretaceous limestone formations, some of which are rich in chert. The parent rock is compact, hard, and decomposes/dissolves slowly, leaving a clayey, loamy, reddish residue that gradually accumulates and is commonly known as terra rosa in the Mediterranean region. The local variety of terra rosa is specific because of the high proportion of siliceous chert material and the relatively high concentrations of some trace elements—HMs. The Calcaric Cambisol soil types in Istria have formed on carbonate flysch formations from the Upper Cretaceous. The flysch in this area typically consists of alternating layers of

quartz sandstone and marlstone of varying thicknesses. The relatively soft flysch weathers more quickly and leaves behind much more silty/sandy fine soil than limestone. These two sedimentary parent materials differ in terms of their physical parameters, and most importantly, they differ in terms of the content of trace elements, especially As, Cd, Ni, and Cr. As already mentioned, the HM concentrations found in the soils are of both natural and anthropogenic origin. Below, we present the distribution of HMs in vineyard soils to try and show, but not prove, the contribution of anthropogenic sources of HMs (mainly viticulture) to the natural concentrations found in soils—this comprises the natural background.

Arsenic

Arsenic (As) concentrations in the Karst vineyards ranged from 13.9–33.2 mg kg$^{-1}$ in the topsoil (0–20 cm) and 13.9–35.3 mg kg$^{-1}$ in the 20–40 cm layer. In the topsoil of the Istrian vineyards, the As concentrations ranged from 4.3 to 27.2 mg kg$^{-1}$ (Figure 2). The average As concentrations were 19.7 mg kg$^{-1}$ in the Karst winegrowing region in the 0–20 cm layer, 20 mg kg$^{-1}$ in the Karst winegrowing region in the 20–40 cm layer, and 13.4 mg kg$^{-1}$ in the Istrian winegrowing region. No sample in either region met or exceeded the As CTV. In the Istria region, no sample met or exceeded the WTV.

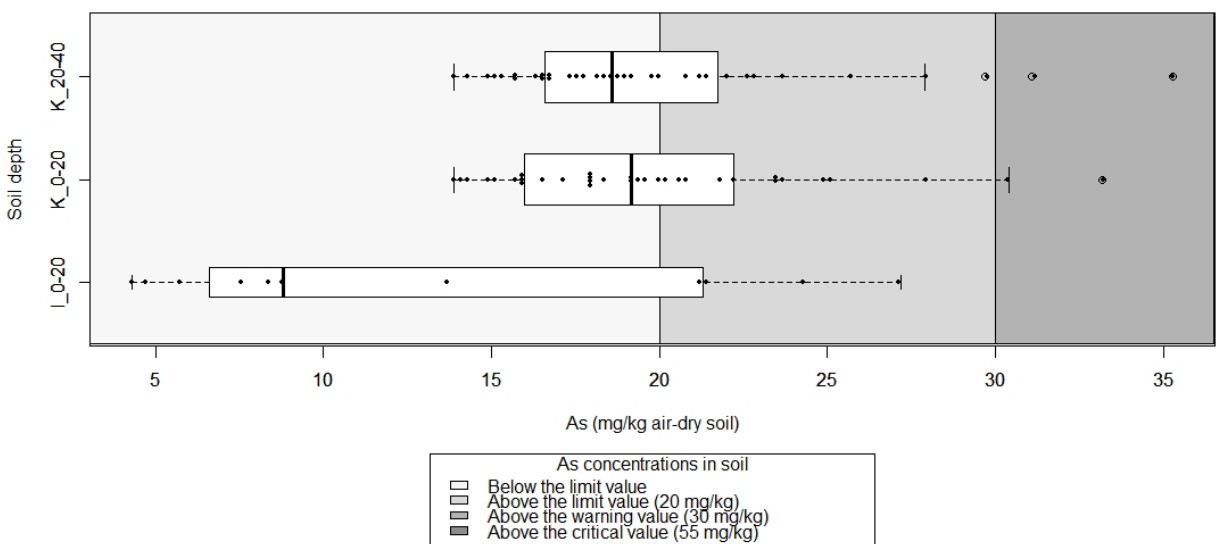

**Figure 2.** As concentrations in Karst vineyard soils (at depths of 0–20 cm and 20–40 cm) and in Istria (0–20 cm) soils in relation to the limit values, as defined by Slovenian legislation [13].

Overall, the large proportion of As concentrations detected at both depths is evenly distributed between a range of very low concentrations which are well below the As CTV (20 mg kg$^{-1}$) (i.e., As concentrations that can be considered close to the natural background, and below the As WTV (30 mg kg$^{-1}$), that is, concentrations that can be considered elevated, and in the opinion of some, concentrations that clearly indicate a significant anthropogenic influence (contamination)). Only one As concentration in the topsoil at one site and As concentrations in the second soil layer at two sites exceeded the As WTV. According to Slovenian legislation (RS, 1969), the soils at these sites are considered contaminated for general agricultural production. The majority of samples of vineyard soils from Istria had much lower As concentrations. The median As concentration was significantly lower (ca. 8.5 mg kg$^{-1}$) than in the Karst Chromic Cambisol soil types. The concentrations of three of the eleven samples that were elevated and exceeded the LTV can be considered the result of anthropogenic activities—primarily viticulture.

The origin of As in European agricultural topsoil is generally considered to be predominantly geological [17]. Nevertheless, Tóth et al. [17] reported that more than 1% of samples

from 15% of EU regions had As concentrations above 50 mg kg$^{-1}$; in seven regions, the number of such samples was above 5%, and in three regions, it reached or exceeded 10%. In selected agricultural areas (mainly France, Italy, and Spain), the highest As concentrations were above 100 mg kg$^{-1}$—a significantly higher concentration than the measurements derived from vineyard soils in Karst (max. 33.2 mg kg$^{-1}$) and Istria (max. 27.2 mg kg$^{-1}$).

Cadmium

Cadmium (Cd) concentrations in the vineyard soil samples were detected in a range between 0.10 and 4.1 mg kg$^{-1}$ in the 0–20 cm layer, and 0.20 and 3.0 mg kg$^{-1}$ in the 20–40 cm layer in the Karst vineyard soils (Figure 3). The average Cd concentration was 0.9 mg kg$^{-1}$ in the soil layer at a depth of 0–20 cm, and 0.8 mg kg$^{-1}$ at a depth of 20–40 cm. The Cd concentration in the topsoil of the vineyards in Istria ranged between 0.16 and 0.58 mg kg$^{-1}$, and it was 0.3 mg kg$^{-1}$ in the lower layer. In Istria, no sample reached or exceeded the LTV and/or WTV. In both regions, no Cd concentration reached or exceeded the CTV (12 mg kg$^{-1}$).

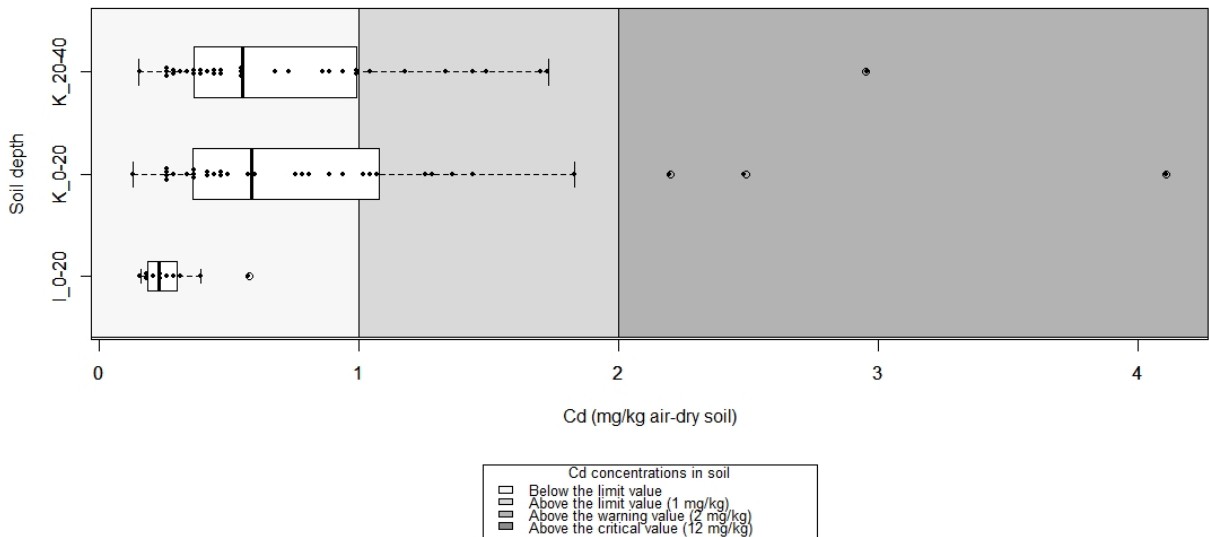

**Figure 3.** Cd concentrations in Karst vineyard soils (at depths of 0–20 cm and 20–40 cm) and in Istria (0–20 cm) in relation to the limit values, as defined by Slovenian legislation [13].

In both regions, Cd concentrations at both depths were very low in a large proportion (62.9%) of the samples. The average concentration was well below the Cd LTV (1 mg kg$^{-1}$). In 22.9% of the samples, the Cd concentrations can be considered elevated, and thus, they may be considered anthropogenic Cd sources. Cd concentrations in the topsoil of two sites, and in the 20–40 cm layer for two sites, exceeded the Cd WTL (2 mg kg$^{-1}$). These sites can therefore be considered contaminated with Cd due to human activities. In contrast to the Karst Chromic Cambisols, the Istrian Calcaric Cambisols that formed on carbonate Flysch formations had significantly lower Cd concentrations. All samples were clearly below the LTV value.

Rusjan et al. [18] observed mean Cd concentrations of 0.9–2.5 mg kg$^{-1}$ in the 0–20 cm layer, and mean concentrations of 1.1–2.5 mg kg$^{-1}$ in the 20–40 cm layer for vineyard soils in the Goriška Brda winegrowing region in Slovenia. Calcaric Cambisol, which developed on the same parent material (carbonate flysch), is the predominant soil type in Goriška Brda and Istria. In the European Union, Cd concentrations in topsoil show a different pattern [17]. Tóth et al. [17] reported that only 5.5% of agricultural soil samples had concentrations above 1 mg kg$^{-1}$; however, in some cases, soils in France and Spain had Cd concentrations above 10 mg kg$^{-1}$ or even above 20 mg kg$^{-1}$, which clearly exceeds the Cd concentrations in

our samples. The results are only consistent to some extent, given the elevated Cd content in the soils of the Burgundy region of France, which are found on Jurassic limestone [19]. Vázques et al. [20] determined average Cd concentrations of 0.439–0.101 mg kg$^{-1}$ in the 0–20 cm layer of vineyard soils in Spain.

Cobalt

The analysed cobalt (Co) concentrations in the topsoil of the vineyards in the Karst region ranged from 12.3–41.0 mg kg$^{-1}$, and 14.9–40.7 mg kg$^{-1}$ in the lower layer. The average Co concentration was 24.1 mg kg$^{-1}$ and 23.8 mg kg$^{-1}$ at depths of 0–20 and 20–40 cm, respectively ( Figure 4). The Co concentrations in the topsoil of the Istrian vineyards ranged from 14.6 to 42.8 mg kg$^{-1}$, whereas the average Co concentration was 22.6 mg kg$^{-1}$.

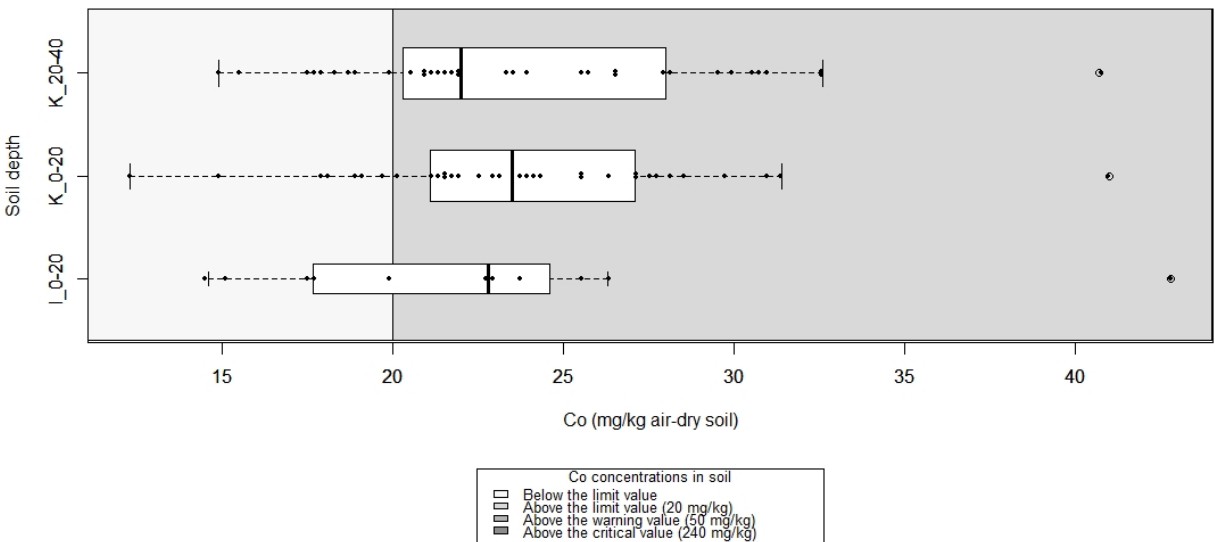

**Figure 4.** Co concentrations in Karst vineyard soils (at depths of 0–20 cm and 20–40 cm) and in Istria (0–20 cm) in relation to the limit values, as defined by Slovenian legislation [13].

The average Co concentration was slightly above the Co WTV (1 mg kg$^{-1}$). Indeed, 80% and 76.5% of the Co concentrations were in the elevated range (20–40 cm and 0–20 cm, respectively). In contrast to the soils of the Karst vineyards, 76.5% of the Co concentrations in the soils of Istria were below the Co WTV. In both regions, the Co concentrations did not exceed the Co CTV. Co concentrations in the Calcaric Cambisol of Istria were lower, but not significantly lower.

Almost identical Co concentrations were found by Rusjan et al. [18], who reported on the vineyard soils of Goriška Brda. The mean Co concentrations in the topsoil were 18–28 mg kg$^{-1}$ for the 0–20 cm layer, and almost the same concentration (18–26 mg kg$^{-1}$) was found in the lower 20–40 cm layer. Cobalt was detected in concentrations higher than 20 mg kg$^{-1}$ for the vineyard soils of Goriška Brda. Cobalt was detected in most European regions at concentrations that exceeded 20 mg kg$^{-1}$. In only one region in France, and in four vineyards in Greece, were concentrations detected that exceeded 100 mg kg$^{-1}$ and 250 mg kg$^{-1}$, respectively [17].

Chromium

Chromium (Cr) concentrations ranged from 70.6–145 mg kg$^{-1}$ (0–20 cm layer) and 49.9–133 mg kg$^{-1}$ (20–40 cm) in the Karst viticulture region (Figure 5). The average Cr concentrations were 94.4 mg kg$^{-1}$ and 93.0 mg kg$^{-1}$ (topsoil and 20–40 cm depth, respectively). Both average values were in a range that was significantly below the LTV value, and thus,

their composition can be considered close to that of the natural background—human influence can be considered limited. In the topsoil of vineyards in Istria, the Cr concentrations ranged from 37.5 to 178 mg kg$^{-1}$ (the average Cr concentration was 86.5 mg kg$^{-1}$). The Cr concentration in only one sample from Istria (and none from the Karst) exceeded the WTV. No sample in either region exceeded the CTV; therefore, the vineyard soils of Istria and Karst are considered uncontaminated. The Cr concentrations in the soil were very close to the natural background levels and only one sample from Karst and one from Istria indicated a significant human Cr contribution.

**Figure 5.** Cr concentrations in Karst vineyard soils (at depths of 0–20 cm and 20–40 cm) and in Istria (0–20 cm) in relation to the limit values, as defined by Slovenian legislation [13].

Cr is quite abundant in most agricultural topsoil in the European Union; 2.7% of samples were above 100 mg kg$^{-1}$ and 1.1% of samples were above 200 mg kg$^{-1}$. In some instances, it even exceeded 300 mg kg$^{-1}$ [17].

Copper

Due to the intensive use of copper-based fungicides in the vineyards, increased copper (Cu) concentrations were expected in the vineyard soils of both regions. In the Karst region, Cu concentrations ranged from 35.0 to 304 mg kg$^{-1}$, and in Karst and Istria, the concentrations ranged from 33.0 to 270 mg kg$^{-1}$, respectively. The average Cu values were 95.3 mg kg$^{-1}$ and 87.3 mg kg$^{-1}$ at depths of 0–20 and 20–40 cm, respectively. The distribution of Cu concentration values (Figure 6) shows that the average concentrations were close to the Cu LTV (60 mg kg$^{-1}$), whereas 31.4% of the Cu concentrations exceeded the WTV, and 2.9% (one sample) exceeded the critical threshold. Cu contamination in the Karst region clearly indicates an anthropogenic hot-spot contamination distribution; a large majority of the sites were not contaminated, whereas about one third of the sites can be considered contaminated, and one may be considered critically contaminated with Cu in terms of general agricultural production.

## Cu in vineyard soils

**Figure 6.** Cu concentrations in Karst vineyard soils (at depths of 0–20 cm and 20–40 cm) and in Istria (0–20 cm) in relation to the limit values, as defined by Slovenian legislation [13].

In Istria, the Cu concentrations in the topsoil ranged from 28.2 to 96.7 mg kg$^{-1}$ and the average CU concentration was 55.6 mg kg$^{-1}$. No sample in the winegrowing region of Istria met or exceeded the WTV or CTV. No outliers indicated a significant anthropogenic enrichment of Cu.

As Cu-containing products are often used in agriculture, especially in pesticides applied in vineyards and orchards, this could be the reason why Cu concentrations above 150 mg kg$^{-1}$ or even 200 mg kg$^{-1}$ have been observed in some regions in France and Italy [17]. Rusjan et al. [18] observed mean Cu concentrations of 51–99 mg kg$^{-1}$ in the 0–20 cm layer, and 51–88 mg kg$^{-1}$ in the 20–40 cm layer of the vineyard soil in the Goriška Brda winegrowing region in Slovenia. Vázques et al. [20] determined mean Cu concentrations of 133–258 mg kg$^{-1}$ in the 0–20 cm layer of vineyard soil in Spain, which was sampled in the spring, as our samples were. Cavani et al. [21] reported that the highest value for Cu in vineyard soil in Italy was 1000 mg kg$^{-1}$, which is three times higher than the maximum value observed in this study in Slovenia. Fernández-Calviño et al. [22] determined Cu concentrations in vineyard soils (the layer from 0–20 cm) on the Iberian peninsula ranging from 25–666 mg kg$^{-1}$. Despite one sample exceeding the CTV value in the Karst region (layer from 0–20 cm), we can conclude that Cu concentrations in Iberian vineyards are often very high, and they can even exceed the maximum observed concentration.

Molybdenum

Molybdenum (Mo) concentrations were not elevated in the vineyard soils of either region. In the Karst region, the range of Mo concentrations in the soil was 2.0–0.8 mg kg$^{-1}$ and 1.8–4.4 mg kg$^{-1}$ (soil depth 0–20 and 20–40 cm, respectively); the average Mo concentrations were 5.2 mg kg$^{-1}$ and 5.2 mg kg$^{-1}$ (soil depth 0–20 and 20–40 cm, respectively). In the topsoil of vineyards in the winegrowing region of Istria, Mo concentrations ranged between 0.1 and 13.4 mg kg$^{-1}$, and the Mo average was 2.9 mg kg$^{-1}$. Mo concentrations did not exceed the CTV in either region. One outlier and six "upper whisker" concentrations indicated anthropogenic hot-spot enrichment of the soil with Mo. Figure 7 shows the distribution of Mo concentrations; the average Mo concentration was well below the LTV. The values from two sites indicate that there was human-induced hot-spot contamination.

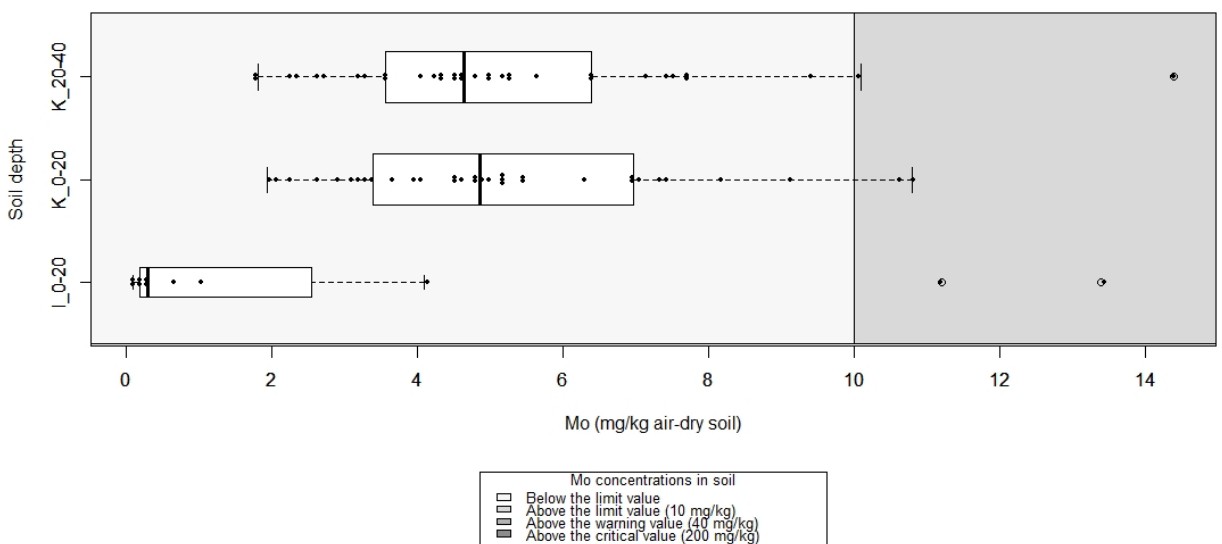

**Figure 7.** Mo concentrations in Karst vineyard soils (at depths of 0–20 cm and 20–40 cm) and in Istria (0–20 cm) in relation to the limit values, as defined by Slovenian legislation [13].

Rusjan et al. [18] observed mean Mo concentrations of 1.1–2.7 mg kg$^{-1}$ in the 0–20 cm layer and 1.4–2.7 mg kg$^{-1}$ in the 20–40 cm layer of vineyard soil in the Goriška Brda winegrowing region in Slovenia.

Nickel

The nickel (Ni) concentrations in the soils of Karst and Istria were expected to be elevated based on some previous measurements that showed a higher natural Ni concentration in the soils of the Primorska region [23]. The Ni concentrations found in this study ranged between 39.7 and 102 mg kg$^{-1}$ and 40.8 and 102 mg kg$^{-1}$ in Karst and Istria, respectively. Figure 8 shows the widely scattered distribution of the measurements in the soils of the vineyards in Karst and Istria. The average values (66.9 and 66.7 mg kg$^{-1}$) for both depths were elevated and slightly below the Ni warning threshold. The almost identical concentrations in the topsoil and lower layer (20–40 cm) confirms, to some extent, the high concentration of Ni in the natural background, and it suggests anthropogenic hot-spot contamination. At elevated Ni concentrations, no sample in either region met or exceeded the CTV for Ni (i.e., no critical contamination was detected).

Tóth et al. [17] concluded that the entire European Union is affected by Ni pollution to some degree. Samples with high Ni concentrations were mainly found in the European Mediterranean, especially in Greece, where concentrations exceeded 100 mg kg$^{-1}$; in some regions, they even reached 150 mg kg$^{-1}$. Compared with the EU–Mediterranean area, Ni concentrations in the vineyard soils of Slovenian Karst and Istria were found to be lower, despite the elevated levels of Ni in the natural background

Lead

Despite some lead (Pb) content and possible traffic-related contamination of the vineyard soils, the Pb concentrations in the vineyard soils of Karst and Istria were found to be low. Pb concentrations in the Karst vineyard soils ranged from 29.1 to 162 mg kg$^{-1}$ in the topsoil, and from 28.9–133 mg kg$^{-1}$ in the lower layer. The average values in both layers were similar (i.e., 43.0 and 40.3 mg kg$^{-1}$ in the topsoil and in the 20–40 cm layer, respectively). Both average values indicated a low anthropogenic influence and a modest natural Pb concentration in the soils. Two Pb measurements in the topsoil, one in the lower layer, and several Pb concentrations in the upper whisker area (Figure 9), are very characteristic of human-induced Pb contamination, and thus Pb hot pot contamination. The situation in

the Istrian vineyards with regard to Pb contamination was similar; indeed, despite even lower median Pb concentrations in the topsoil (26.8 mg kg$^{-1}$), no concentrations exceeded the LTV in Istria—no extreme anthropogenic hotspot contamination was detected.

**Figure 8.** Ni concentrations in Karst vineyard soils (at depths of 0–20 cm and 20–40 cm) and in Istria (0–20 cm) in relation to the limit values, as defined by Slovenian legislation [13].

**Figure 9.** Pb concentrations in the soils of Karst vineyards (at depths of 0–20 cm and 20–40 cm) and in Istria (0–20 cm) in relation to the limit values, as defined by Slovenian legislation [13].

Tóth et al. [16] found that Pb concentrations above 60 mg kg$^{-1}$ were found in Italy, but none of the samples exceeded 200 mg kg$^{-1}$. Rusjan et al. (2006) measured mean Pb concentrations of 19–30 mg kg$^{-1}$ in the 0–20 cm layer and 18–30 mg kg$^{-1}$ in the 20–40 cm layer in vineyard soils in the Goriška Brda winegrowing region in Slovenia. Vázques et al. [20] found average Pb concentrations of 72.6–99.3 mg kg$^{-1}$ in the 0–20 cm layer of vineyard soil in Spain.

Zinc

Zinc (Zn) is the active substance in several vine-protection fungicides. Higher Zn concentrations were therefore expected, especially in the topsoil of vineyards. All Zn concentration values in the topsoil and in the layer between the 20 cm and 40 cm depths did not exceed the LTV for Zn (200 mg kg$^{-1}$). The average Zn concentration values in the karst area were 103 and 98.9 mg kg$^{-1}$ for the topsoil and subsoil, respectively (Figure 10). In accordance with national legislation [13], vineyard soil, which is often publicly declared as being polluted with Zn, was actually shown to be unpolluted, with respect to the presence of Zn.

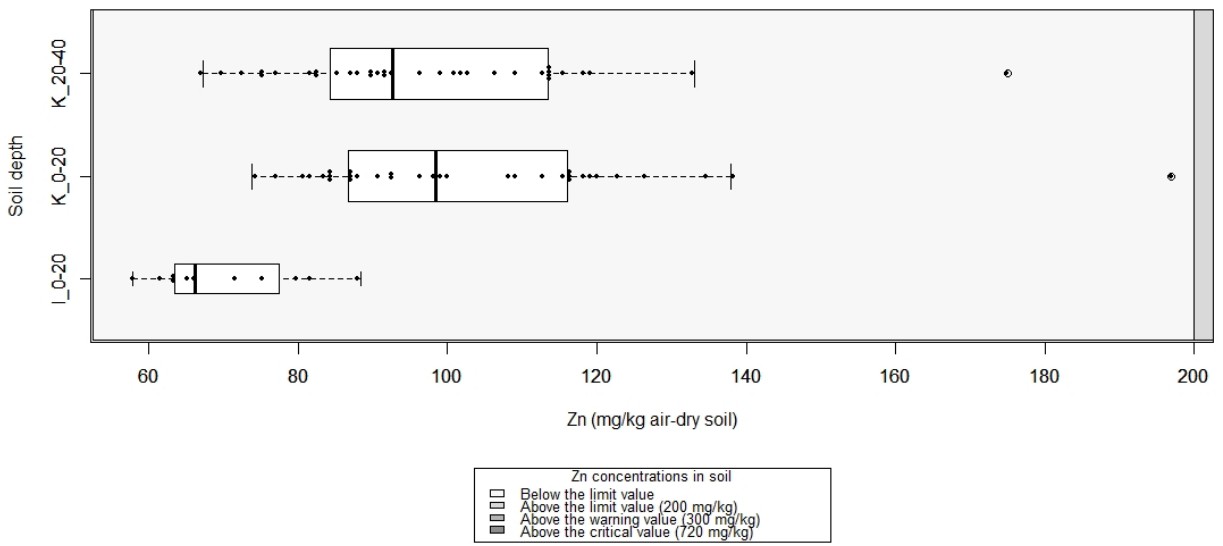

**Figure 10.** Zn concentrations in the soils of Karst vineyards (at depths of 0–20 cm and 20–40 cm) and in Istria (0–20 cm) in relation to the limit values, as defined by Slovenian legislation [13].

Tóth et al. [17] found that Zn concentrations above 200 mg kg$^{-1}$ were found in less than 0.5% of the samples studied. Rusjan et al. [18] measured mean Zn concentrations of 90–118 mg kg$^{-1}$ in the 0–20 cm layer and 87–113 mg kg$^{-1}$ in the 20–40 cm layer of vineyard soil in the Goriška Brda winegrowing region in Slovenia. Vázques et al. [20] measured average Zn concentrations of 98.9–126 mg kg$^{-1}$ in the 0–20 cm layer of vineyard soil in Spain; these samples were taken in the spring, as were our samples.

### 3.4. Discussion

3.4.1. Pesticide Residues in the Vineyard Soils of the Karst and Istria

In the vineyard soils in Karst, the active substances boscalid (a fungicide), chlorothalonil (a fungicide), chlorpyriphos (an insecticide), DDT (an insecticide), dimethomorph (a fungicide), quinoxyfen (a fungicide), and tetraconazole (a fungicide) were found. Only chlorpyriphos (an insecticide) was found in Istrian vineyard soil. The active substances found in the Karst area were obviously due to the use of fungicides in the vineyards. Pesticide residues in the topsoil of Karst (0–20 cm) were higher than in the lower layer (20–40 cm), with the exception of DDT. The measured concentrations of the active substances of pesticides were considered very low. Moreover, 63.8% of the samples from Karst, and 81.8% of the samples from the winegrowing areas of Istria, did not contain residues. The concentrations were ≤ 0.03 mg kg$^{-1}$ dry soil, which, with the exception of the significantly more prevalent DDT, could be due to their rapid degradation on the soil surface, as well as in the soil itself. The maximum pesticide residue concentrations of selected pesticides in some vineyard soils in France [28] and Spain [14,29] were much higher than these for the most part.

### 3.4.2. Heavy Metals in the Vineyard Soils of the Karst and Istria Winegrowing Regions

As concentrations in vineyard soils were low, with about 62% of the measured values being below the LTV, vineyard soils can generally be considered uncontaminated as far as the relatively strict Slovenian legislation is concerned. Certain concentrations exceeded the WTV, and two sites can be considered heavily contaminated with As. In contrast with 15% of the agricultural topsoil in the EU, the measured As concentrations in the vineyard soils of the Slovenian Karst and Istria were significantly lower. In a large portion of the samples (62.9%), the Cd concentrations in the soils of both regions were very low at both depths. Only 22.9% of the measurements exceed the LTV and could be considered elevated. Single sites (2) exceeded the WTL and showed manmade hotspot contamination. The Cd concentrations in the soils of the vineyards in Karst and Istria were comparable with the values determined elsewhere in the EU, including the individual hot-spot concentrations. The average Co concentrations were slightly above the WTV (80% of the topsoil samples and 76% of the subsoil samples showed elevated Co concentrations). The Co concentrations can be compared with the results of the EU and other Slovenian studies. Both the average Cr concentrations in the topsoil and in the second layer were in a range well below the LTV value and can thus be interpreted as being close to the natural background levels. The anthropogenic source of Cr in the soils can be interpreted as limited. Only one Karst and one Istrian site showed Cr concentrations above the WTV value. The Cr concentration is comparable to some findings on EU soils. It was expected that Cu soil concentrations would be elevated in both regions due to the intensive use of Cu-based fungicides in the vineyards. Surprisingly, about 50% of the samples had Cu concentrations below the LTV, and the WTV was exceeded in a small number of samples. Mo concentrations in the soils were low; 94% of the samples from Karst and 81% of the samples from Istria had Mo concentrations below the LTV. The detected Ni concentrations confirm previous studies' findings on HMs; these are published in the journal *Soil Contamination Research* in Slovenia [23]. The concentrations in the topsoil and in the lower layer were similar; the average Ni concentration was close to the WTV (70 mg kg$^{-1}$). Despite the elevated levels of Ni in the natural background, the Ni concentration in vineyard soils in Slovenian Karst and Istria was lower than in the agricultural soils of the EU Mediterranean region [18,22]. The occurrence of Pb in soils was low. The vast majority of measurements (94% and 97% in the topsoil and lower layer in Karst, respectively, and 100% in Istria) were well below the LTV (85 mg kg$^{-1}$) (i.e., the vineyard soils had low natural background levels of Pb, thus indicating marginal human influence). The Zn concentration in the vineyard soils was low, although it is an active ingredient of several fungicides used for pest control in viticulture. All Zn concentration values measured in the topsoil and in the layer at a depth of 20–40 cm did not exceed the LTV for Zn (200 mg kg$^{-1}$).

### 3.4.3. Heavy Metal Concentrations in Grapes and Wine Produced in Istria and Karst

Despite the HM concentrations measured in the soils of vineyards in Karst and Istria, the end products (i.e., grapes and wine) are safe for consumers in terms of HM concentration levels. In the Karst region [24,25] and Istria [26], the HMs in the grapes were below the WTV, and therefore, they may be considered safe for consumers. This is confirmed by analyses of the wine and grapes. Only two of the eighty-two wine samples that were analysed contained Cu levels above the established maximum level. The concentrations of Cu in these two wine samples were 1.03 mg L$^{-1}$ and 1.89 mg L$^{-1}$. The established maximum value is 1 mg L$^{-1}$ [27]. These exceedances are likely due to the excessive use of copper(II) sulphate in wine production. In the Istria region, the HMs in the grapes were below the established maximum levels, and therefore, they are considered to be safe for consumers. Only one of sixty-six wine samples examined contained Zn values above the specified maximum value [26]. The concentration of Zn in that wine sample was 6.14 mg L$^{-1}$ The established maximum value is 5 mg L$^{-1}$ [27].

Nevertheless, nearly all of the grape samples examined (i.e., at least 97.6% of the wine samples examined from the Karst wine region and at least 98.4% of the wine samples

examined from the Istria wine region) can be considered safe for consumers with regard to pesticide residues and HMs.

The concentrations of As, Cd, Co, Cr, Mo, Ni, Pb, and Zn found in vineyard soils in Istria and Karst are below the CTV. Only one soil sample in the Karst vineyard region exceeded the CTV for Cu in the 0–20 cm layer. The higher concentrations of Cu, which were frequently found in the topsoil of the vineyards, were probably due to the long-term use of Cu-based fungicides. The Bordeaux mixture (copper(II) sulphate—$CuSO_4$) has been regularly used in this area for more than 120 years in order to protect vines.

### 3.4.4. General Conclusions

Despite the intensive and often extensive use of pesticides in viticulture, the end products from the Karst and Istria regions (grapes and wine) can be considered safe for consumers in terms of pesticide residues in the soil, the HM contamination of the soil, and HM pesticide residues in the wine. The transition to organic viticulture is justified and cannot be questioned in terms of the potential contamination of the soil with pesticides and HMs. Small winegrowers in Karst and Istria can rightly focus on organic production because their vineyard soils are not burdened by HM pollution and pesticide residues.

**Author Contributions:** B.V. and H.B.Č., conceptualization and writing-original draft preparation; K.L. and S.R., methodology; H.B.Č., Š.V.B., validation; H.B.Č. and Š.V.B., formal analysis; B.V., writing—review and editing. All authors have read and agreed to the published version of the manuscript.

**Funding:** The study was jointly financed by European funds within the Agrotur/Karst Agro-tourism Project (www.agrotur.si, accessed on 1 August 2022) under the Cross-border Cooperation Programme Italy—Slovenia 2007–2013, financed by the European Regional Development Fund and national funds. The research was also financed by the Malvasia TourIstra project (http://www.malvasia-touristra.eu, accessed on 1 August 2022), implemented under the Cross-border Cooperation Programme Croatia—Slovenia 2007–2013, and financed by the European Regional Development Fund and national funds. The authors acknowledge the financial support from the Slovenian Research Agency (Research Core Funding No. P4-0133 and P4-0431) for additional data analysis, interpretation and preparation of the manuscript.

**Data Availability Statement:** The data presented in this study are available on request from the corresponding author. The data are not publicly available due to technical reasons.

**Acknowledgments:** We thank the winemakers of Teran PTP from the Karst winegrowing areas, and the winemakers of Malvazija from the Slovenian and Croatian border region of Istria, for kindly allowing us to take soil samples and for participating in the study. We thank colleagues for their contribution as follows: Marjan Šinkovec and Dejan Bavčar, soil sampling; Mateja Fortuna, Danijela Cvijin, and Marjeta Černe Kanc, formal analysis; Vida Žnidaršič Pongrac, supervision of analytics of heavy metals; Petra Mežič; data visualization.

**Conflicts of Interest:** The authors have declared no conflict of interest.

## Appendix A

The multiresidual GC/MS method was used for the detection of 86 active substances: acrinathrin, aldrin, azinphos-methyl, azoxystrobin, bifenthrin, boscalid, bromopropylate, bupirimate, captan, carbaryl, carbofuran, carboxin, chlorothalonil, chlorpropham, chlorpyriphos, chlorpyriphos-methyl, clomazone, cyhalotrin-lambda, cypermethrin, cyproconazole, cyprodinil, DDT, deltamethrin, diazinon, dichlofluanid, dimethachlor, diniconazole, diphenylamine, endosulphan, endrin, esfenvalerate, fenamidone, fenbuconazole, fenitrothion, fenthion, fenvalerate, flonicamid, fludioxonil, fluquinconazole, folpet, HCH-alpha, HCH beta, HCH-delta, heptachlor, heptenophos, hexachlorobenzene, indoxacarb, iprodione, kresoxim-methyl, lindane, malathion, mecarbam, metalaxyl, metalaxyl-M, methacrifos, methidathion, metrafenone, myclobutanil, oxadixyl, parathion, parathion-methyl, penconazole, permethrin, phorate, phosalone, pirimicarb, pirimiphos-methyl, procymidone, profenofos, propargite, propyzamide, pyridaphenthion, pyrimethanil, quinalphos, quin-

oclamine, quinoxyfen, tebuconazole, tetraconazole, tetradifon, tolclofos-methyl, tolylfluanid, triadimefon, triadimenol, triazophos, trifloxystrobin, and vinclozolin.

The multiresidual LC/MS/MS method was used for the detection of 90 active substances:

3-hydroxycarbofuran, acetamiprid, aldicarb, aldicarb sulphone, azinphosethyl, beflubutamid, benalaxyl, benalaxyl-M, bitertanol, bromuconazole, buprofezin, carbendazim, chlortoluron, clofentezine, clopyralid, clothianidin, cycloxydim, cymoxanil, demeton-S-methyl sulphone, desmedipham, diflufenican, dimethenamid-P, dimethomorph, epoxiconazole, ethofumesate, ethoprophos, etofenprox, famoxadone, fenamiphos, fenarimol, fenhexamid, fenoxaprop-P-ethyl, fenoxycarb, fenpropathrin, fenpropimorf, fenpyroximate, fenthion sulphone, fenthion sulphoxide, fipronil, florasulam, fluazifop-P-butyl, fluazinam, flufenacet, flufenoxuron, fluorochloridone, flusilazole, flutriafol, hexaconazole, hexythiazox, imidacloprid, iprovalicarb, isoproturon, isoxaflutole, linuron, malaoxon, mandipropamid, mepanipyrim, metamitron, metazachlor, methiocarb, methiocarb sulphone, methiocarb sulphoxide, methomyl, methoxyfenozide, metosulam, monocrotophos, napropamide, oxamyl, paraoxon-methyl, pendimethalin, phenmedipham, phorate sulphone, phorate sulphoxide, prochloraz, propaquizafop, pyraclostrobin, pyridaben, pyriproxyfen, spirodiclofen, tebufenozide, teflubenzuron, tebufenpyrad, terbuthylazine, thiacloprid, thiamethoxam, thiodicarb, triasulphuron, trichlorfon, triticonazole, and zoxamide.

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
