# Peer review of "Pesticide Residues and Heavy Metals in Vineyard Soils of the Karst and Istria"

_land, doi:10.3390/land11122332_

Round 1

Reviewer 1 Report

Thank you for the opportunity to read the article entitled “Pesticide residues and heavy metals in vineyard soils of the Karst and Istria” which is an interesting study. The authors seek to determine the levels of pesticide reidues and heavy metals in the topsoil of the samples selected from Slovenian wine-growing region of Karst and of Calcaric Cambisol in 11 vineyards in other Slovenian and Croatian wine-growing regions of Istria. The results are valuable, and the article is mostly well-written. However, I have some reservations which authors must address before it can be accepted.

 The authors need to revise the introduction section to include the importance of vineyard industry in the study areas especially the study regions. Please include some economic statistics, their contribution to the economy and legislation toward sustainable agricultural practices.

 A comprehensive literature review section is missing which encompasses the methodological and regional aspects on the research issues being investigated. In addition to this, a global literature review should be added to make the paper interesting for a wider audience. Authors need to link the paper to sustainable agricultural practices and how the pesticide use is a threat to this. Please included in the literature review the following recent studies along with others:

 https://doi.org/10.1016/j.technovation.2021.102255

https://doi.org/10.3390/ijerph18031175

https://doi.org/10.1016/j.apenergy.2021.118459

In methods section, the authors have not described the sample selection strategy which may have important implications for the overall results of the study. Please explain how the samples were selected and which technique/design was used?

Please rename the Summary to Discussion and contexulaized the findings as well as compare with the extant literature.

Based on your findings, what are your overall recommendations to reduce the use of pesticides and transition toward sustainable agriculture. This is an important question that remains to be answered in the Conclusion section. Please revise.

Author Response

Dear Reviewer,

Thank you very much for your time and for all the comments you've made. We, the authors, are very grateful for your careful reading and suggestions.

The paper has been thoroughly revised and significantly upgraded. Unfortunately, it took some additional time.

The following corrections have been made:

  • The introductory section has been expanded to include the importance of the viticulture industry to small farmers and small wineries (small farms, importance of environmental soil quality to marketing small growers and wineries. Unfortunately, there's no hard data on the economic situation.
  • Thank you for the suggestions to include some studies. After careful consideration, we've concluded that they don't have that much to do with the topic of the paper and therefore could mislead readers. The decision wasn't an easy one.
  • The sampling strategy text was modified and improved, including the sampling technique and sample design (tools, sample locations).
  • The general recommendation to reduce pesticide use is questionable, as national legislation already mandates sustainable and organic agricultural practices.
  • The summary has been renamed to discussion.
  • Also, the English language has been reviewed by a native speaker, a trained professional who specializes in reviewing scientific texts, especially scientific papers.

In addition to your requests, the following corrections/ have been made:

  • An explanation of why the data collected in 2015 weren't published until several years later (unscientific information about highly contaminated soils in vineyards prevails in the public) was added.
  • The tables have been corrected in some places and the legends have been made clearer throughout. Unfortunately, the attempt to present the figures as diagrams wasn't successful because of the very different measurements. We'd to keep the data in tabular form. For the same reason, we decided to keep Table 1, as the information is more readable than if we replace it with two/three charts/diagrams.
  • Figures have been corrected (spelling errors) and recreated.
  • Requested explanation: The paper focuses on soil contamination in relation to past agricultural practices that are more than 150 years old (use of copper for crop protection).
  • Requested explanation: Soil samples were collected in the spring to avoid detection and misinterpretation of possible concentrations of short-lived chemical substances. In fact, the concentrations of some chemical substances used for crop protection decrease significantly after one to two months due to decomposition (UV, T, biota). In practice, the concentration values at the time of completion of the analytical procedures may no longer correspond to the situation in the vineyards.

We thank you again for your time,

Sincerely, the authors.

Reviewer 2 Report

Hi,

1. Improve the abstract, and add the conclusion of the study.

2. Add references in M&M

3. Do not elaborate on the references in the R&D section and try to connect them.

4. In Table 1. define the symbol "/".

5. Instead of tables, figures can be used.

Thanks

Author Response

Dear Reviewer,

Thank you very much for your time and for all the comments you've made. We, the authors, are very grateful for your careful reading and suggestions.

The paper has been thoroughly revised and significantly upgraded. Unfortunately, it took some additional time.

The following corrections have been provided.

  • The abstract was revised, amended and harmonised.
  • An explanation of why the data collected in 2015 weren't published until several years later (unscientific information about highly polluted soils in vineyards prevails among the public).
  • The tables have been corrected in some places and the legends have been clarified throughout. Unfortunately, the attempt to present the figures as graphs wasn't successful due to the wide variation in measurements. We'd to keep the data in tabular form. For the same reason, we decided to keep Table 1 because the information is more readable than if we replaced it with two/three charts.
  • Also, the English language has been reviewed by a native speaker, a trained professional who specializes in reviewing scientific texts, especially scientific papers.

In addition to your requests, the following corrections/ have been made:

  • Page 6 error was corrected.
  • The introductory section has been expanded to include the importance of the viticulture industry to small farmers and small wineries (small farms, importance of environmental soil quality to marketing small growers and wineries. Unfortunately, there's no hard data on the economic situation.
  • The text on sampling strategy has been modified and improved, including sampling technique and sampling design (tools, sampling locations).
  • An explanation of why the data collected in 2015 weren't published until several years later (unscientific information about highly polluted soils in vineyards prevails among the public) was provided.
  • The general recommendation to reduce pesticide use is questionable, as national legislation already mandates sustainable and organic agricultural practices.
  • Explanation to the comment: The paper focuses on soil contamination related to past agricultural practices that are more than 150 years old (use of copper for crop protection).
  • Figures have been corrected (spelling errors) and recreated.
  • Explanation to the comment: Soil samples were collected in spring to avoid detection and misinterpretation of possible concentrations of short-lived chemical substances. In fact, the concentrations of some chemical substances used for crop protection decrease significantly after one to two months due to decomposition (UV, T, biota). In practice, the concentration values may no longer correspond to the situation in the vineyard soils at the time of completion of the analytical procedures.
  • The Summary has been renamed to the Discussion.

We thank you again for your time,

Sincerely, the authors.

Reviewer 3 Report

The article is well written and it reports on the presence of pollutants, in particular pesticides and heavy metals, in the soil of the regions taken from in 69 vineyards at 2 different depths. 

The article is not innovative and the methods are clear but it reports a series of data that can be helpful for other researchers. However, in my opinion, there are some points to define. 

1) Samples have been collected in 2012 and 2015. When Have they been analyzed? The distance in time between the two samples could affect the measurements?

2) why the have been collected only in spring? Some chemicals used in other seasons could be more impacting or dangerous. In addition, a discussion about the evolution of the national regulations in the use of some molecules between 2012-15 and their presence/lack within the two sampling campaigns could be interesting and empower the work.

3) collected data should be presented in a more readable way, for examples by using histograms  (e.g. instead of tables see Table 1)

4) Pg 6 there is an error (Error! Reference source not found)

Author Response

Dear Reviewer,

Thank you very much for your time and for all the comments you've made. We, the authors, are very grateful for your careful reading and suggestions.

The paper has been thoroughly revised and significantly upgraded. Unfortunately, it took some additional time.

The following corrections have been provided.

  • The Abstract and Conclusions were improved, and General conclusions text was added.
  • The symbol “/” was defined in the Table 1.
  • The tables have been corrected in some places and the legends have been clarified throughout. Unfortunately, the attempt to present the figures as graphs wasn't successful due to the wide variation in measurements. We'd to keep the data in tabular form. For the same reason, we decided to keep Table 1 because the information is more readable than if we replaced it with two/three charts.
  • Also, the English language has been reviewed by a native speaker, a trained professional who specializes in reviewing scientific texts, especially scientific papers.

In addition to your requests, the following corrections/ have been made:

  • The text on sampling strategy has been modified and improved, including sampling technique and sampling design (tools, sampling locations)
  • An explanation of why the data collected in 2015 weren't published until several years later (unscientific information about highly polluted soils in vineyards prevails among the public) was provided.
  • Figures have been corrected (spelling errors) and recreated.
  • The summary has been renamed to discussion.
  • Requested explanation: The paper focuses on soil contamination related to past agricultural practices that are more than 150 years old (use of copper for crop protection).
  • Requested explanation: Soil samples were collected in spring to avoid detection and misinterpretation of possible concentrations of short-lived chemical substances. In fact, the concentrations of some chemical substances used for crop protection decrease significantly after one to two months due to decomposition (UV, T, biota). In practice, the concentration values may no longer correspond to the situation in the vineyard soils at the time of completion of the analytical procedures.

We thank you again for your time,

Sincerely, the authors.

Round 2

Reviewer 1 Report

After careful evaluation of the revised manuscript, I believe that authors have addressed all the concerns outlined in my earlier review of the manuscript. Therefore, I am pleased to recommend the publication of the article.

Reviewer 2 Report

All the best.